# Recent Advances in Functional Fiber-Based Wearable Triboelectric Nanogenerators

**DOI:** 10.3390/nano13192718

**Published:** 2023-10-06

**Authors:** Hakjeong Kim, Dinh Cong Nguyen, Thien Trung Luu, Zhengbing Ding, Zong-Hong Lin, Dukhyun Choi

**Affiliations:** 1School of Mechanical Engineering, Sungkyunkwan University, Suwon 16419, Republic of Korea; 2Department of Biomedical Engineering, National Taiwan University, Taipei 10167, Taiwan; 3Department of Future Energy Engineering, Sungkyunkwan University, Suwon 16419, Republic of Korea; 4Institute of Energy Science & Technology (SIEST), Sungkyunkwan University, Suwon 16419, Republic of Korea

**Keywords:** triboelectric nanogenerator, functional fiber, wearable electronics

## Abstract

The quality of human life has improved thanks to the rapid development of wearable electronics. Previously, bulk structures were usually selected for the fabrication of high performance electronics, but these are not suitable for wearable electronics due to mobility limitations and comfortability. Fibrous material-based triboelectric nanogenerators (TENGs) can provide power to wearable electronics due to their advantages such as light weight, flexibility, stretchability, wearability, etc. In this work, various fiber materials, multiple fabrication methods, and fundamentals of TENGs are described. Moreover, recent advances in functional fiber-based wearable TENGs are introduced. Furthermore, the challenges to functional fiber-based TENGs are discussed, and possible solutions are suggested. Finally, the use of TENGs in hybrid devices is introduced for a broader introduction of fiber-based energy harvesting technologies.

## 1. Introduction

Recent advances in technology have improved the quality of life around the world. Such technologies can be found in homes, offices, factories, and vehicles. The need for human intervention has been minimized through automation, often using the internet of things (IoT) and wireless sensor networks (WSN) [1]. The fourth industrial revolution accelerated the development of IoT technology, sometimes changing lives dramatically. Personal computers, smartphones, and wearable electronics are connected through the IoT and can be used for practical applications such as smart homes, smart traffic, wearable electronics, and healthcare monitoring [2,3,4,5]. In an IoT network, mobility and wireless power supplies are required because data must be transmitted without temporal or spatial restrictions. Batteries are widely used to power IoT devices, but several challenges remain to be resolved as batteries need to be replaced or recharged periodically. In addition, battery waste is environmental pollution [6,7,8,9]. An alternative energy source is needed to minimize the environmental impact of these technologies.

There is an urgent need for a sustainable energy source due to environmental pollution and climate change induced by rising carbon emissions. Many studies have investigated potential sustainable power sources based on energy harvesting technologies. Energy harvesters produce electricity through triboelectric, piezoelectric, thermoelectric, or electromagnetic effects, etc. [10,11,12,13]. Among the available energy harvesters, triboelectric nanogenerators (TENGs) are popular due to their ease of fabrication, simple working mechanisms, low manufacturing costs, minimal restrictions in selecting materials or structures, and applicability to diverse applications [14,15,16,17,18,19,20,21,22]. The commercialization of TENG is also underway [23]. TENGs can generate electricity from various sources such as wind, ocean waves, acoustic energy, biomechanical energy, etc. [24,25,26,27,28,29,30]. In addition, various materials and fabrication methods can be used for the output power and practical applications [31,32,33,34,35]. Although TENGs offer multiple advantages and can be used in various ways, bulk structures are not suitable for use in portable devices or wearable electronics.

The development of wearable electronics has made our lives more convenient. The requirements for wearable electronics are light weight, flexibility, stretchability, etc. Previously, bulk structures were usually selected for the fabrication of high performance electronics, but they are not suitable for wearable electronics [36]. Fibrous materials can be used to manufacture wearable electronics for practical utilization. A lot of studies have been reported, but there are several challenges for functional fiber-based TENGs. Firstly, long-term stability is one of the limitations of fiber-based TENGs, and it can be improved by the selection of wear-resistant materials. Secondly, wearability is another problem for functional fiber-based TENG, and it can be handled by using biocompatible materials [37]. Furthermore, the energy conversion efficiency of functional fiber-based TENGs is still low, but it can be improved by enhancing the surface modification, chemical treatment, enlarging surface contact area, etc. [38].

Since 1764, fiber-based materials, beginning with natural fibers, have become indispensable [39]. Today, fibrous materials are used in energy harvesting, healthcare monitoring, drug delivery, and wearable electronics, etc. [40,41,42,43]. Several methods of manufacturing fibrous materials have been developed, including coating, spinning, plating, printing, injecting, etc. Fabrication processes and fiber materials are well organized by Prof. Wang’s group, and they will be briefly introduced here (Figure 1) [44]. Fiber-based electronics can be fabricated using coating techniques including dip coating, flow coating, spin coating, roll coating, etc. Dip coating is suitable for coating all surfaces of the materials, however spin coating and casting methods are used to make the different surfaces. The size and shape can be restricted when the dip coating method is applied. Spin coating is usually used with flat and symmetrical substrates. Irregular surfaces can be obtained from the coating method, and proper manufacturing techniques should be chosen to fabricate the fiber materials based on coating techniques according to the purpose of use. Spinning is widely used for the production of fiber materials. Dry spinning is appropriate process when the solvent is easily evaporated, but wet spinning should be considered if the solvent is slow to evaporate. Melt spinning is used when the polymer material is easily melted by the temperature. The plating method is used for the deposition of metals on a surface. A thin layer of metal is accumulated on the surface of the other metals. The plating metal serves as an anode, and the other metal serves as a cathode. Metal ions are transferred by an electrical current provided from an external power source based on the principle of electrolysis. However, metal ions are deposited on the surface with an additional electric field in electroless plating based on the principle of redox reaction. The printing method is used for large-scale fabrication, and it is a cost-effective fabrication process.

Various conductive materials are used for fiber-based nanogenerators including metals, polymers, carbonaceous fillers, liquid electrodes, and hybrid fillers (Figure 1) [44]. Metals are widely used in multiple ways due to their advantages such as high electrical conductivity and excellent mechanical properties. However, it is difficult to use metals for large-scale applications due to their disadvantages, including heavy weight and large volume. Recently, metals have been used in wearable electronics in the form of nanoparticles (NPs), nanowires (NWs), nanosheets, etc. Especially, a lot of metal nanowires including copper nanowires (Cu NWs), gold nanowires (Au NWs), and silver nanowires (Ag NWs) have been reported due to their high electrical conductivity, thermal conductivity, optical properties, etc. Moreover, conductive polymers are used in various fields due to their light weight, corrosion resistance, excellent mechanical properties, etc. There have been a lot of conductive polymers including polyaniline (PANI), polypyrrole (PPy), polythiophene (PTh), etc. Especially, poly(3,4-ethylenedioxythiophene):polystyrene sulfonate (PEDOT:PSS) is promising material due to its great thermal stability, environmental stability, high electrical conductivity, etc. Furthermore, a lot of carbonaceous fillers have been reported including carbon black (CB), carbon nanotubes (CNT), graphene oxide (GO), etc. Among the carbonaceous fillers, CNTs and graphene are widely used in wearable electronics as electrode materials. Materials and fabrication processes must be appropriately selected based on their purpose and applications.

Various fibrous materials and functional fibers have been studied for practical applications such as acoustic, tactile, and motion sensors, gas detection, and physiological monitoring [45]. Smart fiber- and textile-based TENGs (f/t-TENGs) have been summarized and discussed by Prof. Wang’s group [46]. TENGs are used for energy generation and self-powered monitoring systems, and f/t-TENGs are used in the same way. For a higher output performance in f/t-TENGs, functional materials can be embedded in fibrous materials [47,48,49,50]. In addition, functional fiber-based TENGs are now used in biomedical applications such as pulse wave, respiration, and human motion monitoring, as well as quantitative analysis [51,52,53,54]. Here, we review recent advances in functional fiber-based wearable triboelectric nanogenerators (FF-WTENGs) in the context of healthcare applications, environmental monitoring, sterilization systems, and human–machine interface (HMI) applications (Figure 2).

## 2. Triboelectric Nanogenerator

Static electricity is ubiquitous but is often perceived as problematic rather than useful. However, power can be extracted from static electricity through energy harvesting technologies. Since their introduction in 2012, TENGs have been studied in various ways [64]. Four fundamental modes of TENG are known: the single-electrode (SE) mode, lateral-sliding (LS) mode, vertical contact-separation (CS) mode, and freestanding triboelectric layer (FT) mode (Figure 3) [44]. Firstly, in the single electrode mode, the reference electrode is connected to the ground, and it is widely used in fiber- or textile-based TENGs. Secondly, two tribo-layers perpendicular to the tribo-surfaces are required for the vertical contact-separation mode. However, two tribo-layers parallel to the interfaces are needed in the lateral sliding mode. Furthermore, the triboelectric layer and electrodes are used in the freestanding triboelectric layer mode.

The working mechanism of TENGs can be explained in terms of contact electrification and electrostatic induction in the vertical contact and separation modes. In a triboelectric series, when two dissimilar materials come into contact, the surfaces acquire positive and negative charges due to contact electrification. When the two materials are separated, the charged surfaces return to their original states. Electrostatic induction results from the separation of the two materials. The electrical potential between the two materials creates an electric field, driving free electrons through the electrodes connected to an external circuit. An alternating current can be generated through the periodic contact and separation of the two materials. The working mechanism of TENGs can be explained by the Gauss theorem [65]. The voltage–charge–motion (V-Q-x) relationship is a function of time. In this model, two triboelectric materials and two electrodes are used, as shown in the center of Figure 2. Dielectric material 1 (D1) and dielectric material 2 (D2) are attached to metal electrode 1 (M1) and metal electrode 2 (M2), respectively. Their respective thicknesses are *d*_1_ and *d*_2_, and their dielectric constants are *ε*_*r*1_ and *ε*_*r*2_. When the dielectric materials (D1 and D2) come into contact due to an external force, the inner surfaces are charged with a density of *σ_sc_*. Afterward, an electrical potential (*V*) between the two electrodes is produced by the separation of the two materials through the release of external force. The number of transferred charges between the two electrodes is denoted as *Q*. The voltage difference between the two electrodes can be described considering the air gap, two dielectrics, and the Gauss theorem, as shown in Equation (1):(1)$$V\left(t\right)=E_{1}d_{1}+E_{2}d_{2}+E_{air}x$$

Substituting *σ_sc_* into Equation (1) produces the V-Q-x relationship, as shown in Equation (2).
(2)$$V\left(t\right)=-\frac{Q}{S\varepsilon_{0}}\left(\frac{d_{1}}{\varepsilon_{r1}}+\frac{d_{2}}{\varepsilon_{r2}}+x\left(t\right)\right)+\frac{\sigma_{sc}}{\varepsilon_{0}}x(t)$$

In an open circuit, the current becomes zero as there is no charge to transfer through the electrodes. The open-circuit voltage (*V_oc_*) can be determined by Equation (3).
(3)$$V_{oc}\left(t\right)=\frac{\sigma_{sc}}{\varepsilon_{0}}x(t)$$

In a short circuit, the transferred charges *Q_sc_* can be derived as the voltage potential, as shown in Equation (4).
(4)$$Q_{sc}=\left(\;\frac{{S\sigma }_{sc}x\left(t\right)}{\left(\frac{d_{1}}{\varepsilon_{r1}}+\frac{d_{2}}{\varepsilon_{r2}}+x\left(t\right)\right)}\right)$$

Finally, the short-circuit current (*I_sc_*) can be derived from the preceding equation, as shown in Equation (5).
(5)$$I_{sc}=\frac{dQ_{sc}}{dt}=\frac{d}{dt}\left(\frac{{S\sigma }_{sc}x\left(t\right)}{\left(\frac{d_{1}}{\varepsilon_{r1}}+\frac{d_{2}}{\varepsilon_{r2}}+x\left(t\right)\right)}\right)=\frac{S\sigma_{sc}\left(\frac{d_{1}}{\varepsilon_{r1}}+\frac{d_{2}}{\varepsilon_{r2}}\right)v(t)}{{\left(\frac{d_{1}}{\varepsilon_{r1}}+\frac{d_{2}}{\varepsilon_{r2}}+x\left(t\right)\right)}^{2}}$$

Studies on the fundamental theory of TENGs are in progress. Recently, the working principle of TENGs was explained by Prof. Wang using an expanded Maxwell equation [66]. TENG output performance can be optimized by controlling the key parameters based on an understanding of the working principle from Equation (5) [67].

A triboelectric series is an indicator of the tendency to gain or lose electrons, as proposed by Johan Carl Wilcke in 1757 [23]. The material with greater electron affinity on its surface will acquire a negative charge after contact. Material selection based on the triboelectric series is an important determinant of TENG output performance, as described by Prof. Nie’s group [68]. Recently, interest has grown in the study of liquid–solid and solid–solid TENGs. A liquid triboelectric series was established for the first time by Prof. Kim’s group [69]. As shown in Equation (5), TENG output can be improved by enhancing the surface charge density and selecting the appropriate material and fabrication process. Currently, TENGs are used widely in intelligent applications such as smart agriculture, smart industries, smart cities, and emergency monitoring [70]. It is anticipated that additional applications can be developed by exploiting the advantage of fiber-based materials. Functional fiber-based wearable triboelectric nanogenerators (FF-WTENGs) are introduced in this paper.

## 3. Healthcare Applications

With the advent of the fourth industrial revolution and the rapid growth of IoT technologies, wearable electronics have become popular. Wearable devices can be used for real-time healthcare monitoring by incorporating physical or chemical biosensors [71], although adequate power supplies remain a challenge for further market penetration [72]. Batteries are used widely for wireless power supply, but the need for periodic battery replacement and the environmental pollution associated with used batteries contribute to the need for an alternative power source. Electrical energy can be produced from waste biomechanical energy using TENGs, which can be a sustainable solution for wearable electronics. TENGs that harvest biomechanical energy can be attached to various body parts, such as the head, arm, foot, knee, wrist, and finger [44]. Wearable TENGs can also be integrated with IoT systems for biomedical applications, HMI, and automation applications [73]. Recently, several studies have explored the possibilities of wearable electronics based on smart fibers or smart textiles.

TENGs can be used as self-powered systems in healthcare applications that exploit biomechanical energy sources such as heartbeats, respiration, and human motion [74]. The intended meaning has been retained. Wearable and implantable healthcare applications based on TENGs have been the subject of multiple studies. Physiological signal monitoring can be performed based on an all-textile structured TENG composed of two triboelectric layers of a polyvinylidene difluoride (PVDF)/MXene nanofiber membrane as a tribo-negative layer and Ag@nylon 6,6 fibrous membrane as a tribo-positive layer (Figure 4a) [55]. The nanofiber membranes were prepared via the electrospinning method. The Fourier transform infrared (FT-IR) spectrum of MXene-doped PVDF (P/M) nanofibers is shown (Figure 4b) for the analysis of β-phase content, which can be calculated using Equation (6) [55].
(6)$$C_{\beta }=\frac{X_{\beta }}{X_{\alpha }+X_{\beta }}=\frac{Abr_{\beta }}{\left[\left(\frac{k_{\beta }}{k_{\alpha }}\right)Abr_{\alpha }+Abr_{\beta }\right]}$$
where *Abr_β_* is the absorption intensity at vibration bands typical for *β*-phase, *Abr_α_* is absorption intensity for *α*-phase, *k_α_* is the absorption factor at wavenumber 6.1 × 10^4^ cm^2^/mol, and *k_β_* is the absorption factor at wavenumber 7.7 × 10^4^ cm^2^/mol. The optimized MXene content in the PVDF/MXene nanofiber membrane plays a crucial role in optimizing the TENG output, which can be analyzed via Fourier transform infrared and X-ray diffraction instruments (Figure 4c). Ag@nylon 6,6 nanofibers demonstrated considerable antibacterial activities against Escherichia coli and Staphylococcus aureus due to the presence of silver nanoparticles [55]. An integrated system was used as a wearable sensor for pulse monitoring and as a smart keyboard when attached to human skin. 

Respiration occurs in all living humans and contains useful information about the human body. Compared with blood analysis, respiration monitoring offers advantages such as rapid results, non-invasive techniques, and convenience [75]. Pulse and respiration monitoring can be performed simultaneously through a wearable textile-based TENG [76]. A triboelectric all-textile sensor array (TATSA) can be used for real-time physiological signal monitoring when attached to the neck, wrist, or fingertip. Conductive and commercial nylon yarns are used for manufacturing, and a full cardigan stitch is selected due to its large acting area. To fabricate conductive yarn, stainless steel fiber is used, and several Terylene yarns are twisted around a core fiber to manufacture yarn in the desired color due to the characteristics of commercial nylon yarns. A TATSA exhibits a pressure sensitivity of 7.84 mV/Pa, a rapid response time of 20 ms, long-term stability (more than 100,000 cycles), a wide working frequency bandwidth (up to 20 Hz), and machine washability (more than 40 times). Furthermore, a TATSA can be used to monitor sleep quality or disorders using pulse and respiratory signals.

Technologies that function underwater are essential for national security and marine engineering [77]. Although wearable electronics can be used in underwater environments, several issues need to be addressed, including waterproofing and sustainable power sources [78]. It is possible to develop a TENG capable of healthcare monitoring even in an underwater environment based on functional fibers [79]. Nickel metal–organic frameworks (Ni-MOFs) and PVDFs are used for tribo-negative materials, and the β-phase content of PVDF is increased by the addition of Ni-MOF particles due to the interaction between carboxyl groups of the particles and the difluoromethylene groups of the polymer. In addition, nylon-66 nanofibers are used for the tribo-positive material, and a thin layer of polydimethylsiloxane (PDMS) is encapsulated to provide waterproof properties. When the output was measured to confirm the waterproof property, results outside and inside a water tank were nearly identical. The measured output voltage and current under single-finger tapping were 45 V and 0.77 μA, respectively. Moreover, arterial pulse monitoring and wireless underwater communication have been demonstrated. Monitoring of the movement of scuba divers is also possible with this system.

Wound healing, while essential for personalized healthcare management, is a complex and dynamic physiological process [80]. Nonhealing wounds can lead to complications such as metabolic disorders, immune imbalances, and even life-threatening diseases. Patch-type applications for wound healing are used but are limited by power supplies or antibiotics. A self-powered wound-healing patch can be fabricated with a functional fiber-based TENG (Figure 5a) [56]. Polycaprolactone (PCL) and poly(lactic-co-glycolic acid) membranes can be prepared for a wound healing patch via electrospinning. Polypyrrole (PPY) is coated onto another electrospun PCL membrane via chemical vapor deposition. Then, three membranes are assembled into a sandwich structure device, which functions as a single-electrode-mode TENG. This has exhibited outstanding antibacterial properties and satisfied the physical requirements for a wound-healing patch, including flexibility, breathability, and wettability. Electrical stimulation from the TENGs and positive charges on the PPY surface effectively kill bacteria. Moreover, the antibacterial property can be explained in terms of the electrostatic interaction between PPY molecules and bacterial cells. The in vivo performance was evaluated by healing infected wounds in diabetic rats. The wound-healing TENG patch achieved superior healing compared with control groups (Figure 5b), and the patch showed excellent wound-healing performance compared to the control groups, completely healing up to 14 days after surgery (Figure 5c).

## 4. Environmental Monitoring

The importance of environmental monitoring has been highlighted due to air and water pollution. Practical chemical sensors need to be investigated, and multiple studies have been reported [81]. Moreover, monitoring of the daily ambient environment is required for safety. Temperature is an essential factor in determining the quality of human life, and a variety of temperature sensors have been described. Recently, a self-powered fire alarm electronic textile (SFA e-textile) has been reported, as shown in Figure 6a [57]. A flame-retardant property is required for reliable warnings, and test results indicate an outstanding flame-retardant performance compared with commercial cotton fabric, as shown in Figure 6b. SFA e-textiles are manufactured through wet spinning, sol-gel, freeze drying, spray coating, and weaving (Figure 6c). Sodium alginate, Fe_3_O_4_ nanoparticles (NPs), and CaCl_2_ are used to fabricate hydrogel fibers. Ethylene glycol was used for densely structured hydrogel fibers through the solvent exchange process. Ammonium polyphosphate and silver nanowires (AgNWs) were then spray-coated onto the fiber surface to fabricate a thermal-induced conductive aerogel fiber (TIC-AF). It was then freeze dried to make it ultralight. Finally, the SFA e-textile was manufactured through a weaving process. The TENG output has been optimized according to the fiber diameter and AgNW spraying time. The changes in electrical resistance in response to temperature can be used reversibly at temperatures between 100 and 400 °C. In the fire warning test, Fe_3_O_4_ NPs play a crucial role. At high temperatures, electrical resistance changes rapidly, which enables a rapid alarm response in high fire-risk situations. Furthermore, the TENG exhibited excellent antibacterial properties due to the spraying of AgNWs on SFA e-textile. When released, silver ions penetrated the cytomembrane, resulting in the killing of bacteria.

Severe accidents often occur in hazardous environments such as high-temperature fire environments or toxic chemical environments. Appropriate protection or real-time monitoring of temperature or chemical enviroment help prevent these accidents. However, flame retardant properties or pH resistance is required for practical environmental monitoring. Recently, poly(m-phenylene isophtalamide) fiber and a carbon nanofiber composite triboelectric nanogenerator (PMIA/CNF-TENG) were reported for risk perception in fire and chemical environments [82]. PMIA nanofibers were prepared via electrospinning. SiO_2_-modified carbon nanofiber (SiO_2_@CNF) nonwoven fabrics were prepared via electrospinning, pre-oxidization, and carbonization. The PMIA/CNF-TENG was fabricated by painting a PEDOT:PSS aqueous dispersion onto a single face of a CNF nonwoven fabric. Afterward, PMIA nonwoven fabric was adhered to the coated surface of the CNF fabric. Finally, PMIA/CNF-TENG was treated with 1H, 1H, 2H, 2H-perfluorooctyltriethoxysilane to make it resistant to strong acidic and alkaline environments. The PMIA/CNF-TENG exhibited great thermal stability and flame retardance, as confirmed via thermogravimetric measurement and a vertical burning test. Its performance in flame retardant tests in terms of heat release rate, total heat release curve, time to ignition, and total consumed oxygen was outstanding. Furthermore, the highest V_0_ grade for the flame retardant test was achieved with PMIA/CNF-TENG. Excellent chemical resistance is required for PMIA/CNF-TENG as it is used for hazardous environment monitoring in strongly acidic or alkaline environments, and it was grafted with fluorine polymer to enhance its chemical resistance. It is stable in a hazardous chemical environment because there is no large change in properties when exposed to acidic and alkaline solutions, and chemical monitoring was possible due to the change in TENG output according to pH value.

The early detection of dangerous situations and the prevention of accidents are essential for safety in our daily lives and on industrial sites. In this regard, a fabric triboelectric nanogenerator (F-TENG) for self-powered high-risk environment monitoring has been reported, as shown in Figure 7a [58]. The F-TENG achieved excellent chemical resistance when exposed to sulfuric acid compared with conventional fabrics (Figure 7b) and can be used for self-powered chemical leakage monitoring, as shown in Figure 7c. Moreover, it can monitor the working environment using vital signs and enable real-time remote alarms. Figure 7d indicates that a smart, chemical-resistant, high-risk environmental monitoring system has good washability and chemical resistance. Excellent washability was confirmed by the absence of a dramatic change in TENG output after five washes. When exposed to acid and alkali solutions, the output tends to increase because the specific surface area increases as the solutions etch the surface of the PTFE filaments. Furthermore, F-TENG shows excellent hydrophobic properties compared to cotton fabric, as shown in Figure 7e.

## 5. Sterilization Systems

Although wearable electronics have made life more convenient, prolonged use increases the probability of a bacterial infection of human skin. Although the sterilization of wearable electronics has not received much attention, it is essential for their healthier use. Recently, a self-powered sterilization system with a biocompatible nano/microporous fiber triboelectric nanogenerator (NMF-TENG) has been reported, as shown in Figure 8a [59]. A composite nanofiber prepared with PA-66 and ethyl cellulose (EC) based on electrospinning is used as the tribo-positive material. A PTFE membrane was prepared with PTFE microporous film, PTFE substrate, and a conductive silver layer. These two tribo-materials were assembled for a vertical contact-separation mode TENG structure. Nanowire interdigital electrode films were prepared from a carbon nanotube (CNT), N-methylpyrrolidone (NMP), indium tin oxide (ITO), and silver nanowires (AgNWs). Afterward, the end of the AgNW/CNT/ITO nanowire interdigital electrodes were connected to the two terminals of a nano- and microfiber TENG with wires. The optimized ratio between PA66 nylon and EC was chosen based on TENG output performance and mechanical properties as a tribo-positive material (Figure 8b). A high voltage output (302 V) for sterilization was obtained even at a working frequency of 1 Hz, as shown in Figure 8c. Self-powered sterilization is composed of two parts: a power module and an electrode module. The power module is an NMF-TENG that generates electric energy from low-frequency mechanical motion. The electrode module is composed of AgNW/CNT/ITO nanowire interdigital electrodes and is integrated into a power module through external leads, where it achieved excellent antibacterial properties compared with control groups. Silver ions can be released from AgNWs in high-voltage electric field conditions and adsorbed to the cell membrane. Self-sterilization is achieved by reactions between silver ions and cell wall peptidoglycan, which results in the destruction of unique components of bacteria and disordered production.

Multifunctional wearable electronics and TENG-based wearable electronics can be applied to various functions. For example, an ultraviolet (UV)-protective, self-cleaning, antibacterial nanofiber-based TENG has been reported, as shown in Figure 9a [60]. Titanium dioxide (TiO_2_) is known for its UV resistance, stability, and lack of toxicity. However, the content of TiO_2_ was selected based on UV transmittance, absorption, and its protective performance and mechanical properties, as shown in Figure 9b. TiO_2_ and polyacrylonitrile (PAN) can be used to fabricate nanofibers, which provide UV protection and electrification (Figure 9c). Thermoplastic polyurethane (TPU) and AgNWs are used to fabricate other nanofibers and serve as an electrode. PTFE is sputtered onto the middle film via magnetron sputtering technology for enhanced hydrophobicity. A multifunctional wearable TENG is composed of three parts: a TiO_2_@PAN nanofiber, PTFE, and TPU/AgNWs. Three fabrication procedures were compared in terms of UV protection: electrospinning PAN nanofibers and electrospraying TiO_2_ (TiO_2_/PAN); electrospinning PAN and coating with a pre-dispersed TiO_2_ NPs immersion (TiO_2_-PAN); and electrospinning of a solution based on TiO_2_ and PAN (TiO_2_ + PAN). As a result, TiO_2_ + PAN nanofibers from one-step electrospinning achieved the best UV protective properties considering the UV protection factor, transmittance of UVA, and transmittance of UVB. Silver and TiO_2_ are anti-microbial agents with a wide range of biocidal properties. When the anti-bacterial performance was evaluated, TiO_2_/Ag NW/PAN nanofibers demonstrated the best anti-bacterial properties compared with the control groups due to the free radicals from TiO_2_ and rapidly released silver ions. Furthermore, TiO_2_ can be used for self-cleaning under UV light as it is a semiconductor material. The electrical output recovery was evaluated when polluted by organic contaminants (Figure 9d). Samples containing TiO_2_ recovered to near-original values.

## 6. Human–Machine Interface Application

There has been tremendous progress in human–machine interface (HMI) technology due to the rapid development of IoT technology and fifth generation (5G) telecommunications. HMIs make it possible to interact with computers and robots more conveniently in various fields such as healthcare monitoring, industrial applications, and smart home applications, etc. [83]. However, they suffer from high power consumption and a bulky structure due to the signal collection and processing steps. Self-powered HMIs can be operated continuously without an external power supply, and it is possible to manufacture flexible and miniaturized wearable electronics using TENGs and fibrous materials. For example, a multifunctional self-powered haptic sensor with a humidity-resistant, stretchable, and wearable textile-based TENG has been reported [61]. A porous flexible layer (PFL) was prepared for attaining a superhydrophobic property based on the sacrificial template method, and a waterproof flexible conductive fabric (WFCF) was prepared via multi-layered chemical deposition (Figure 10a). The PFL was used as a tribo-negative layer and the WFCF as a conductive electrode. The textile-based TENG (PFL@WFCF-TENG) achieved high output due to the three-dimensional structure of the PFL and the superior conductivity of the WFCF. Moreover, it showed excellent humidity-resistance and stretchability (Figure 10b). Furthermore, the recovery time for energy generation was reduced to 120.6 s (Figure 10c). Finally, a multifunctional self-powered wearable haptic controller has been demonstrated based on various HMI scenarios: controlling the switch of a lamp, electronic badges, computer applications, and the operation of a humidifier (Figure 10d). The trigger signal is produced after rectification and power smoothing following the generation of an original signal using human touch based on a PFL@WFCF-TENG-based haptic sensor.

Outdoor accidents can lead to severe injury or even death, and requesting a rescue at the right time can mitigate dangerous situations. Wearable or portable devices can be used to request a rescue in an emergency, and it is possible to manufacture battery-free, self-powered wearable electronics using a fabric-based TENG [62]. A bionic-scale knitting triboelectric nanogenerator (BSK-TENG) can be fabricated using a knitting technique based on a high-speed V-bed flat knitting machine with three types of yarns (Figure 11a). PTFE yarn is used as a tribo-negative material that is hydrophobic. Nylon yarn is a widely used textile materials due to its strong abrasion resistance and low price. Silver-plated nylon yarn is used as a conductive material due to its soft feel and low electrical resistance. A BSK-TENG operates in a single-electrode mode, and the electrical performance can be optimized with knitting parameters (vertical scales, horizontal distance between scales, width and length of scales, and scale layout). The mechanical properties, washability, breathability, and durability of BSK-TENGs have been investigated. When the electrical output was evaluated after washing in a laundry machine, the output performance recovered after one day (Figure 11b). Finally, integrating a BSK-TENG as a self-powered sensor with Bluetooth low-energy communication allows for practical applications (Figure 11c) in outdoor rescue systems with a wireless signal transmission (Figure 11d).

Electronic skin is a powerful technology for the monitoring of physiological signals and human movement. Moreover, it can be used in multiple ways in connection to HMI technologies. Recently, bio-inspired directional moisture-wicking electronic skin (DMWES) has been reported (Figure 12a) [63]. PVDF, carboxylated carbon nanotubes (CNTs), polyacrylonitrile (PAN), and MXene were used for fabrication through electrospinning and electrospraying. Carboxylic CNT–modified PVDF (C-PVDF) nanofibers were manufactured via electrospinning on aluminum foil. Afterward, MXene/CNT conductive ink was coated onto C-PVDF nanofibers via electrospraying. Finally, PAN nanofibers were electrospun onto C-PVDF/MXene-CNTs as a superhydrophilic outer protective layer. DMWES can be used for effective biomechanical energy harvesting and precise bioelectric signal sensing due to its unique structures. Figure 12b depicts the change in water contact angle over time. Only the C-PVDF layer achieved stable hydrophobicity, and this property was used for directional water transport. The electronic skin demonstrated a rapid response time (28.4 ms) and recovery time (39.1 ms) due to the characteristics of the C-PVDF layer (Figure 12c). Such a skin can be used for gait monitoring based on the single-electrode mode due to its pressure-sensing abilities. Precise human-pulse monitoring is also possible.

## 7. Hybrid System Based on TENG

Triboelectric nanogenerators (TENGs) are promising energy harvesters and used in various fields. Moreover, TENGs can be utilized more widely by forming a hybrid system with other energy harvesters. There have been a lot of TENG-based hybrid devices for the enhancement of power density, utilization in various conditions, the use of various sources, etc. [84]. TENG-based hybrid harvesting systems are well organized by Prof. Cao’s group with respect to devices and applications (Figure 13) [85]. TENGs can be integrated with other energy harvesters such as a piezoelectric generator, electromagnetic generator, thermoelectric generator, etc. [86,87,88,89,90].

A lot of studies have been reported regarding the principle and various applications of piezoelectric nanogenerators (PENGs) ever since their first introduction in 2006 [91]. The piezoelectric potential difference generated by strain cause the electron flows in PENG [92]. Particularly, PENGs require piezoelectric materials such as polyvinylidene fluoride (PVDF), zinc oxide (ZnO), lead zirconate titanate (PZT), etc. Although PENGs have limitations in their material selection, enhanced energy conversion efficiency can be expected through hybrid systems with TENGs.

Electromagnetic generators (EMGs) are energy harvesters based on the electromagnetic induction effect, which was discovered by Michael Faraday in 1831. EMGs exhibit a high energy conversion efficiency in the range of high frequency, and the induced electrodynamic potential can be described as in Equation (7) [93].
(7)$$E=n\;\cdot \;\frac{∆\Phi }{\;∆t}$$
where *n* is the turns of coil, and ΔΦ/Δt is the change rate of magnetic flux in each coil.

The output performance of a TENG and EMG hybrid system can be optimized considering the impedance, working frequency, and electrical properties. 

There have been a lot of studies related to the hybrid system based on TENG and a thermoelectric generator (TEG) ever since its first demonstration in 2013 [94]. The figure of merit ZT is the crucial parameter for the evaluation of thermoelectric performance, and it can be described as in Equation (8) [95].
(8)$$ZT=\frac{\alpha^{2}}{k}\sigma T$$
where α is the Seebeck coefficient, σ is the electrical conductivity, *k* is the thermal conductivity, and *T* is the temperature. Mechanical energy and thermal energy can be the energy sources for the TENG and TEG hybrid system. In this way, hybrid systems with other energy harvesters allow TENGs to be utilized more efficiently.

## 8. Conclusions

Since their first introduction in 2012, the fundamental theory, tribo-materials, structural design, circuits, and applications of TENGs have been studied extensively. A variety of materials and structures can be used in TENGs, but fiber-based TENGs are more suitable than bulk structures for utilization as wearable electronics. In this work, recent advances in functional fiber-based wearable TENGs are introduced. Firstly, various fiber materials and manufacturing procedures are introduced in this work. It is necessary to select fiber materials and fabrication processes according to the environments in which they are used. Moreover, the fundamentals of TENGs were explained for a better understanding of TENGs. Furthermore, functional fiber-based TENGs have been categorized and described in terms of healthcare applications, environmental monitoring, sterilization systems, and human–machine interfaces. Lastly, hybrid systems based on TENGs are introduced for a broader introduction to energy harvesting technologies. Recently, the roadmap for TENGs has suggested movement toward commercialization [23]. Continuous progress has been made toward the commercialization of TENGs, and it is expected that commercialization will be furthered by the development of functional fiber-based TENGs for safe and convenient human life.

## Figures and Tables

**Figure 1 nanomaterials-13-02718-f001:**
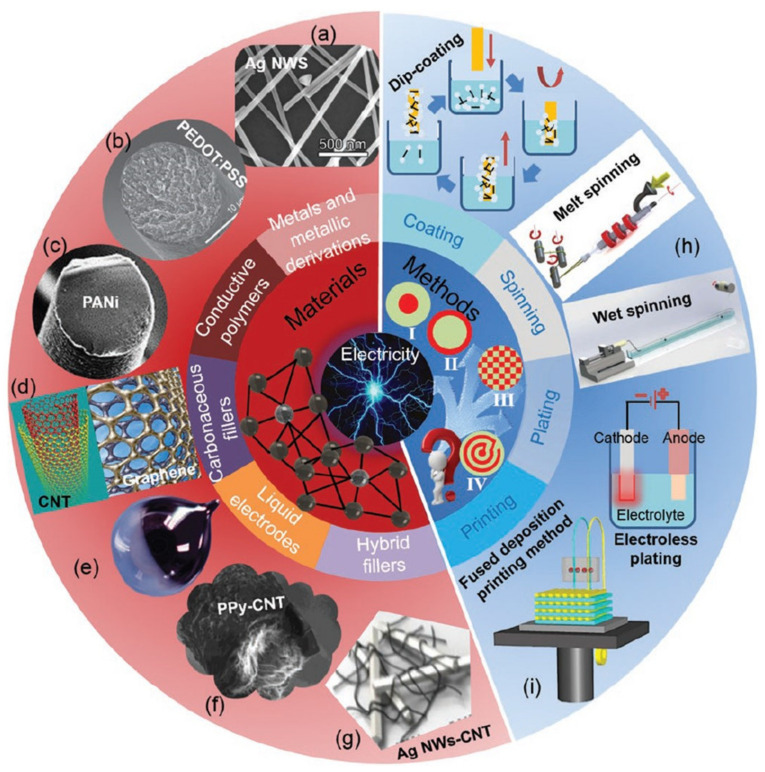
Commonly adopted conductive material systems and fabrication methods of electronic textiles. The frequently used conductive materials for textiles are broadly divided into five categories, i.e., metals and metallic derivations, conductive polymers, carbonaceous fillers, liquid electrodes, and their hybrid fillers. (**a**) Ag NWs. (**b**) PEDOT:PSS. (**c**) PANI. (**d**) CNT and graphene. (**e**) liquid metal. (**f**) PPy−CNT. (**g**) Ag NWs−CNT. The main preparation methods for conductive textile materials include coating, spinning, plating, and printing. There are four main types of compounding structures between conductive materials and other functional materials, i.e., inner embedding (I), outer covering (II), homogenous blending (III), and spiral cladding (IV). (**h**) Melt spinning and wet spinning. (**i**) Fused deposition printing. Reprinted with permission from Ref [44]. Copyright 2019 John Wiley & Sons.

**Figure 2 nanomaterials-13-02718-f002:**
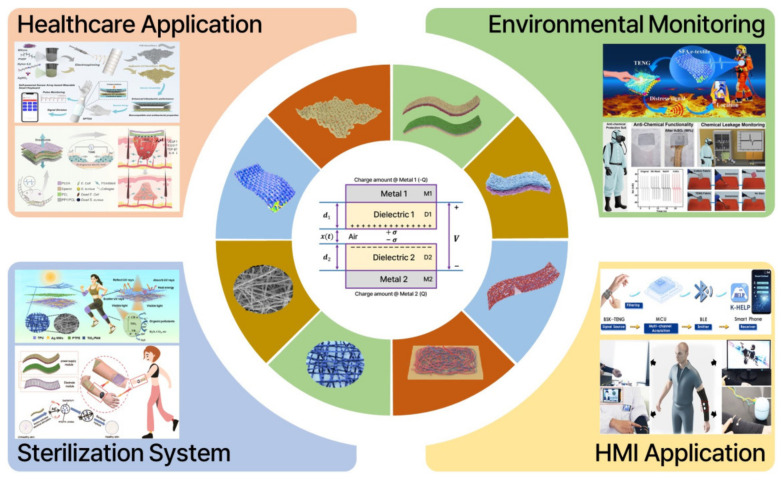
Functional fiber−based wearable triboelectric nanogenerators for use in healthcare applications, environmental monitoring, sterilization systems, and human−machine interfaces. Reprinted with permission from Ref. [55]. Copyright 2023 Elsevier. Reprinted with permission from Ref. [56]. Copyright 2023 American Chemical Society. Reprinted with permission from Ref. [57]. Copyright 2022 American Chemical Society. Reprinted with permission from Ref. [58]. Copyright 2021 John Wiley & Sons. Reprinted with permission from Ref. [59]. Copyright 2023 American Chemical Society. Reprinted with permission from Ref. [60]. Copyright 2021 American Chemical Society. Reprinted with permission from Ref. [61]. Copyright 2021 Elsevier. Reprinted with permission from Ref. [62]. Copyright 2022 Elsevier. Reprinted with permission from Ref. [63]. Copyright 2023 Springer Nature.

**Figure 3 nanomaterials-13-02718-f003:**
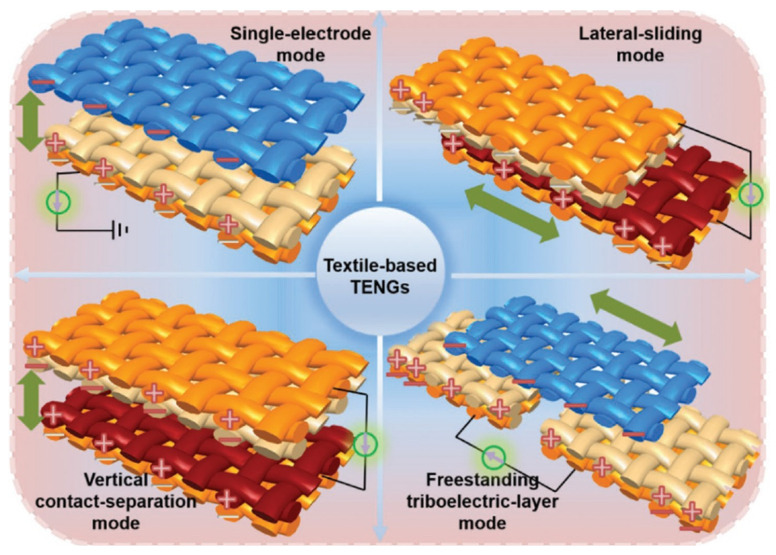
The four operation modes of textile−based TENGs, which take the fabric−based TENGs as examples. The yellow models refer to conductive fabrics, while the gray, crimson, and blue models represent dielectric fabrics. Reprinted with permission from Ref. [44]. Copyright 2019 John Wiley & Sons.

**Figure 4 nanomaterials-13-02718-f004:**
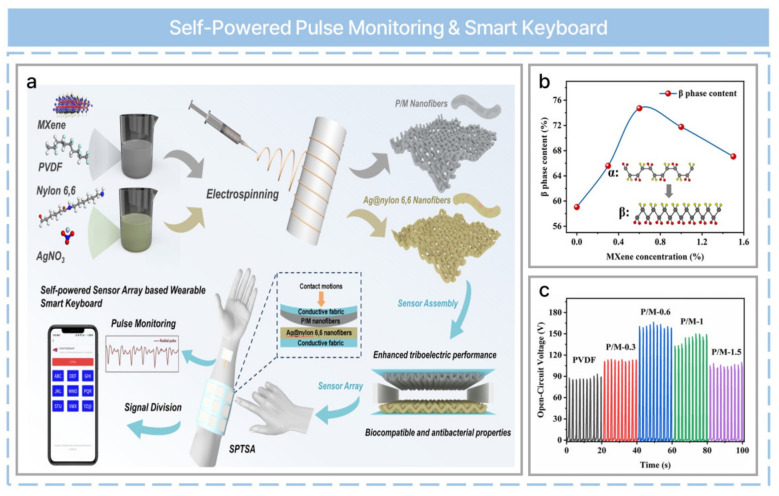
A biocompatible and antibacterial all−textile structured triboelectric nanogenerator for a self−powered tactile sensing apparatus (SPTSA). (**a**) Schematic of the fabrication and application of the SPTSA. (**b**) The β-phase content of the P/M nanofibers. (**c**) The open-circuit voltage variations of different P/M nanofibers. Reprinted with permission from Ref. [55]. Copyright 2023 Elsevier.

**Figure 5 nanomaterials-13-02718-f005:**
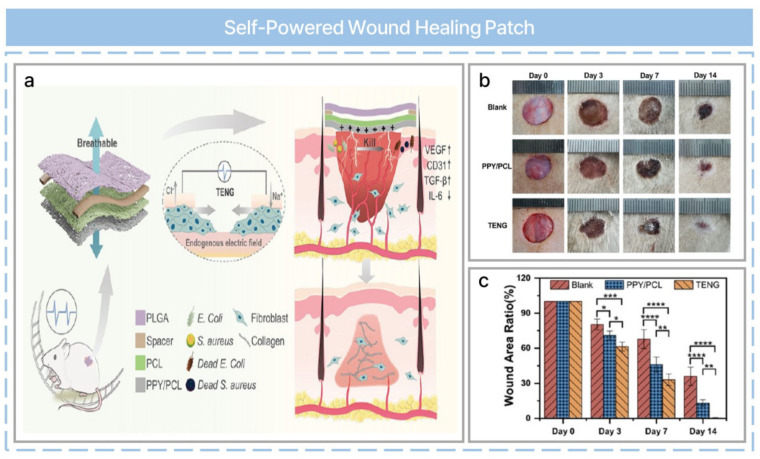
Flexible, breathable, and self-powered patch assembled from electrospun polymer triboelectric layers and a polypyrrole-coated electrode for infected chronic wound healing. (**a**) Schematic illustration of self−powered multifunctional wound−healing patch. (**b**) Photographs of wounds at different periods. (**c**) Wound-area statistics at different periods. All experiments were conducted with at least three sample sizes, and the experimental results were expressed as mean ± standard deviation. Differences between groups were analyzed by one−way analysis of variance (ANOVA), with *p*-values less than 0.05 considered statistically significant (* *p* < 0.05, ** *p* < 0.01, *** *p* < 0.001, and **** *p* < 0.0001). Reprinted with permission from Ref. [56]. Copyright 2023 American Chemical Society.

**Figure 6 nanomaterials-13-02718-f006:**
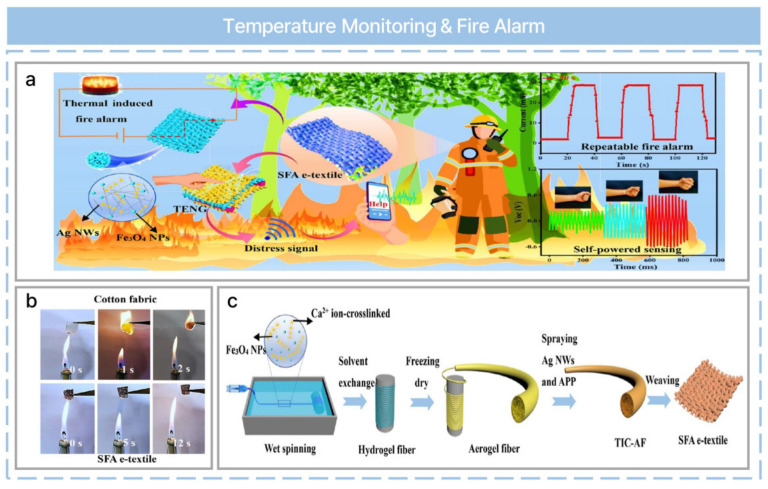
Ultralight self−powered fire alarm e−textile based on a conductive aerogel fiber with repeatable temperature monitoring performance used in firefighting clothing. (**a**) Applications of the SFA e−textile in smart firefighting clothing for energy harvesting, real−time fire warning, and precise rescue location. (**b**) Vertical burning test processes of the SFA e−textile under an alcohol lamp flame. (**c**) Schematic description of the fabrication of the TIC−AF and SFA e−textile. Reprinted with permission from Ref. [57]. Copyright 2022 American Chemical Society.

**Figure 7 nanomaterials-13-02718-f007:**
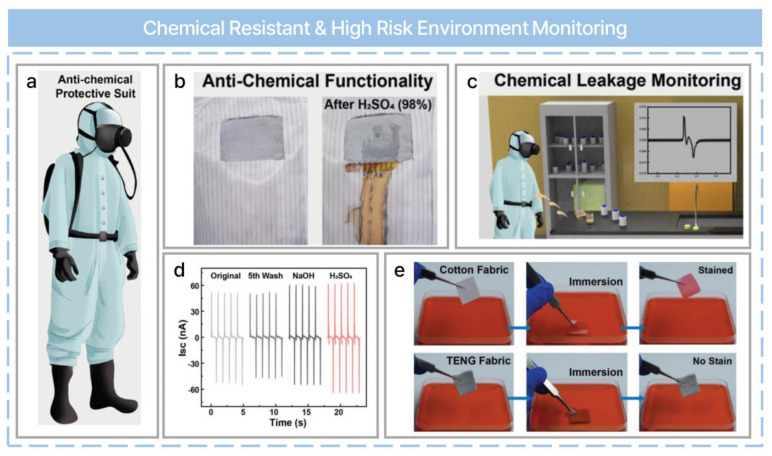
Acid− and alkali−resistant textile triboelectric nanogenerator as a smart protective suit for liquid energy harvesting and self−powered monitoring in high−risk environments. (**a**) Schematic of an anti-chemical protective suit. (**b**) Anti-chemical functionality. (**c**) Chemical leakage monitoring. (**d**) Electrical output performance after water washing and acid and alkali soaking. (**e**) Photographs showing the dipping of F−TENG and cotton fabric into colored water. Reprinted with permission from Ref. [58]. Copyright 2021 John Wiley & Sons.

**Figure 8 nanomaterials-13-02718-f008:**
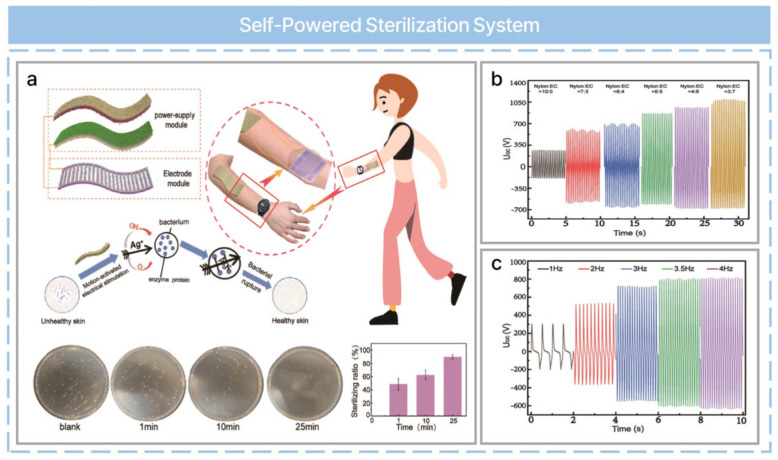
Self−powered sterilization system for wearable devices based on biocompatible materials and a triboelectric nanogenerator. (**a**) Schematic of motion−activated self−powered sterilization. (**b**) Open−circuit voltage of NMF-TENGs constructed from PA66/EC nanofiber membranes with different mass ratios. (**c**) Open−circuit voltage of NMF-TENGs under different impact frequencies at PA66 and EC at the best quality (PA66/EC = 5:5). Reprinted with permission from Ref [59]. Copyright 2023 American Chemical Society.

**Figure 9 nanomaterials-13-02718-f009:**
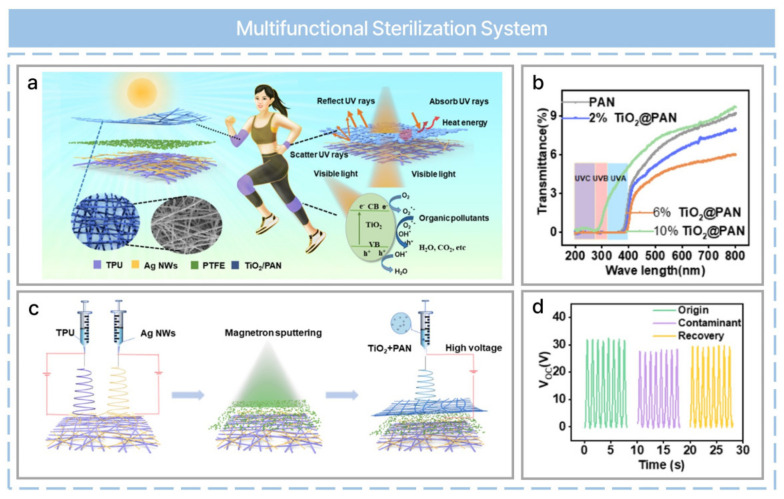
UV−protective, self−cleaning, and antibacterial nanofiber−based triboelectric nanogenerators for self-powered human motion monitoring. (**a**) Schematic of a TENG. (**b**) UV transmittance spectra of nanofibers at different TiO_2_ concentrations. (**c**) An illustration of the fabrication process of the TiO_2_ + PAN nanofibers. (**d**) Comparison of *V_oc_* among the original, polluted, and self−cleaned TENGs. Reprinted with permission from Ref. [60]. Copyright 2021 American Chemical Society.

**Figure 10 nanomaterials-13-02718-f010:**
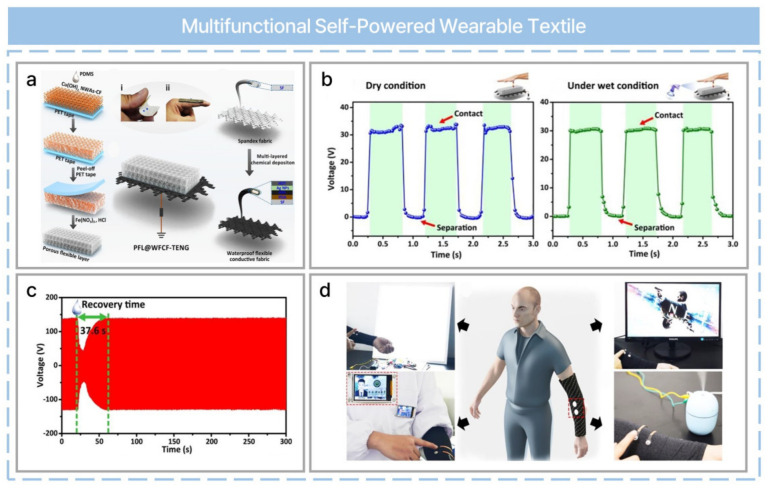
A humidity−resistant, stretchable, and wearable textile−based triboelectric nanogenerator for mechanical energy harvesting and multifunctional self−powered haptic sensing. (**a**) Fabrication process of PFL@WFCF−TENG. (**b**) Comparison of the sensing performance of PFL@WFCF−TENG on finger touching between dry and wet conditions. (**c**) The output signal recovery process curve of the t-TENG after spray dampness adopts triboelectric layers’ PFL. (**d**) Schematic multi−functional HMI systems based on wearable haptic controllers such as controlling light switch states, electronic badges, slides, and humidifiers. Reprinted with permission from Ref. [61]. Copyright 2021 Elsevier.

**Figure 11 nanomaterials-13-02718-f011:**
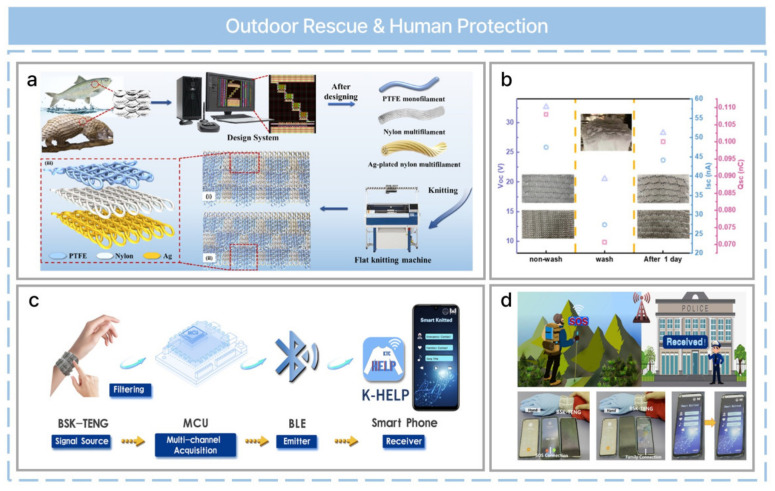
Industrial production of a bionic-scale knitting fabric-based triboelectric nanogenerator for outdoor rescue and human protection. (**a**) Structural design and fabrication of a BSK−TENG. (**b**) The output performance of BSK−TENGs before washing, after washing, and one day after washing. (**c**) A schematic of a BSK−TENG as a self-powered sensor for a wireless personal outdoor rescue system. (**d**) Demonstration of the personal outdoor rescue system with wireless signal transmission using a BSK-TENG as a self−powered sensor and specific functions of the personal outdoor rescue system. Reprinted with permission from Ref. [62]. Copyright 2022 Elsevier.

**Figure 12 nanomaterials-13-02718-f012:**
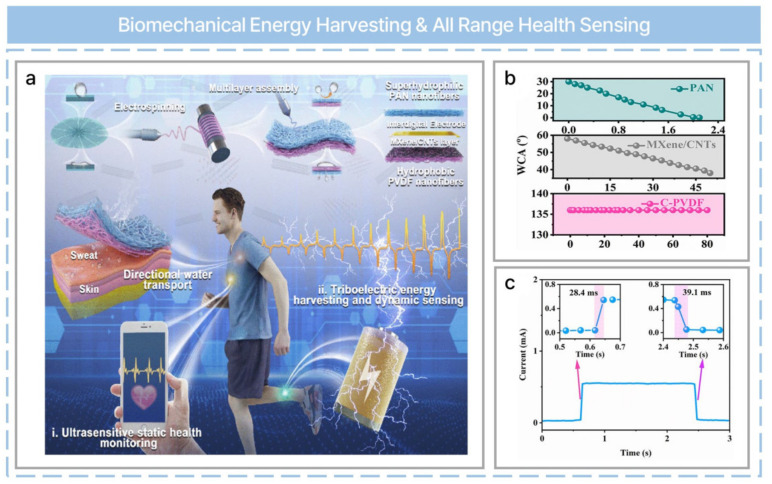
Bioinspired all−fibrous directional moisture−wicking electronic skins for biomechanical energy harvesting and all−range health sensing. (**a**) Schematic of the fabrication and application of the DMWES membrane. (**b**) Water contact angle change with time. (**c**) Response and recovery time of the DMWES. Reprinted with permission from Ref. [63]. Copyright 2023 Springer Nature.

**Figure 13 nanomaterials-13-02718-f013:**
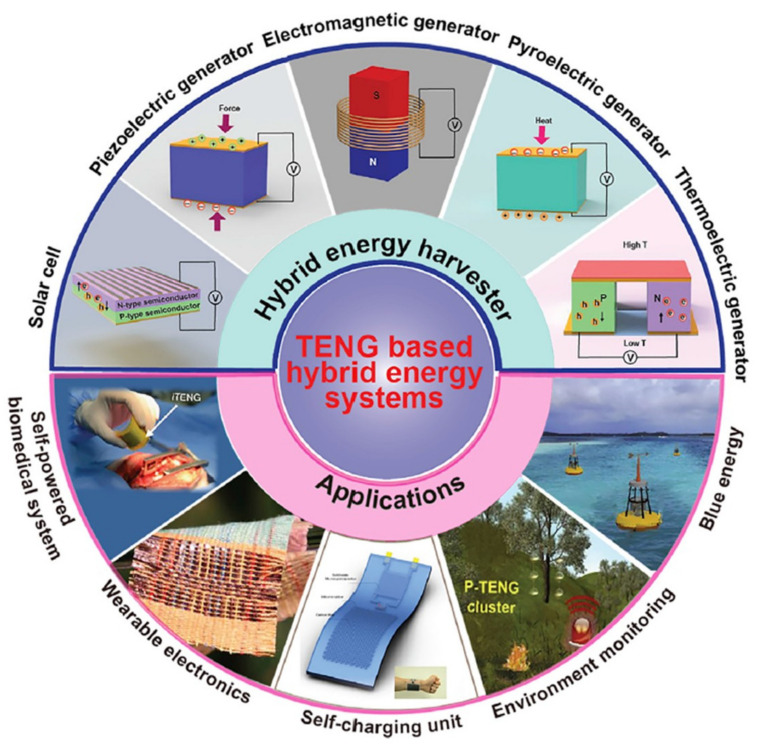
Outline illustration of hybrid energy harvesting systems using triboelectric nanogenerators. The hybrid energy harvesters are integrating triboelectric nanogenerators (TENGs) with other major energy harvesting techniques including electromagnetic generators, piezoelectric generators, thermoelectric generators, pyroelectric generators, and solar cells. They can be used for a variety of applications such as a self−charging power system, self-powered biomedical system, wearable electronics, environment monitoring, and wave energy harvesting. Reprinted with permission from Ref. [85]. Copyright 2021 Elsevier.

## Data Availability

No new data were created or analyzed in this study. Data sharing is not applicable to this article.

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
