# Peer review of "Recent Advances in Functional Fiber-Based Wearable Triboelectric Nanogenerators"

_nanomaterials, 2023, doi:10.3390/nano13192718_

Round 1
Reviewer 1 Report
The figures are hard to read. I like to see more of the construction and data. I think it is very interesting topic. There are a couple of spelling mistakes - please revise.
Needs to be revised for minor spelling and word choices.
Reviewer 2 Report
This review relates to the manuscript ID “2610566” submitted to "nanomaterials" journal entitled, “Recent Advances in Functional Fiber based Wearable Triboelectric Nanogenerators". The authors review fiber-based triboelectric nanogenerators for wearable applications based on the literature survey. The abstract is reasonable as it covers the aspects discussed. The comments are given below.
- The present review is interesting as discussed here the theory of TENG, but it would be great if they can add towards industrial applications.
- There should be a motivation part of these technologies and the importance/disadvantages of fiber based wearable triboelectric nanogenerators for better reading and visibilities for the article.
- The authors need copyright permission for the published Figures and add in the captions. Double-check all the figures.
- The authors may add some recent literature that may be added to the manuscript to make the introduction more interesting and comprehensive. e.g., Advanced Functional Materials 30 (48), 2004446, 2020, Advanced Energy Materials 9 (9), 1803027, ACS Applied Energy Materials 1 (9), 4963-4975, 2018, etc.
- There reviewer suggests adding a section with different materials based fibers used in TENG for the future researcher so that they have excitement about the TENG energy harvesting technology.
The authors should make minor edits to the English language
Reviewer 3 Report
Please see the attachment.

Reviewer 4 Report
The article describes the functional fibre-based triboelectric materials for energy harvesting applications in several key sectors, primarily healthcare and human-machine interfaces. In my opinion, there are plenty of review articles on fibre-based TENG. This manuscript is nothing new in that respect. However, if the paper is organized in a different format, it could offer more to the potential readers. Several points need to be addressed before consideration for publication:
a. the abstract is incomplete and insufficient with the work presented, rather full of background information that has been described in introduction. More emphasis on fibre is important, why its superior to other materials.
Reorganize and rewrite the abstract.
b. Electronic skin has stronger references with respect to human-machine interaction, considering it in aspect of healthcare only narrows the scope of this field. Please reorganize this part.
c. section headings are not clearly justified: the subsections are more like application oriented. If that's the case, how come theory of TENG placed at beginning?
d. Section 3 is too big, splitting it in several subsections will make it more attractive to readers. Please reconsider.
e. Several key references have to be incorporated to validate the article:
Nano Energy 115 (2023) 108707; Materials Horizons 2022, 9 (5), 1468-1478
English language seems fine, except some spelling mistakes. A thorough check before revised submission is advised.
Round 2
Reviewer 4 Report
Manuscript has no issues now. Ready to be accepted. Congratulation